# Indirect Economic Impact Incurred by Haze Pollution: An Econometric and Input–Output Joint Model

**DOI:** 10.3390/ijerph16132328

**Published:** 2019-07-01

**Authors:** Jibo Chen, Keyao Chen, Guizhi Wang, Rongrong Chen, Xiaodong Liu, Guo Wei

**Affiliations:** 1School of Mathematics and Statistics, Nanjing University of Information Science & Technology, Nanjing 210044, China; 2National Climate Center, China Meteorological Administration, Beijing 100081, China; 3School of Computing, Edinburgh Napier University, Edinburgh EH10 5DT, UK; 4Department of Mathematics and Computer Science, University of North Carolina at Pembroke, Pembroke, NC 28372, USA

**Keywords:** Econometric (EC) model, input–output (IO) model, static and dynamic EC + IO joint models, haze pollution, indirect economic loss

## Abstract

Econometrics and input–output models have been presented to construct a joint model (i.e., an EC + IO model) in the paper, which is characterized by incorporating the uncertainty of the real economy with the detailed departmental classification structure, as well as adding recovery period variables in the joint model to make the model dynamic. By designing and implementing a static model, it is estimated that the indirect economic loss for the transportation sector caused by representative haze pollution of Beijing in 2013 was 23.7 million yuan. The industrial-related indirect losses due to the direct economic losses incurred by haze pollution reached 102 million yuan. With the constructed dynamic model, the cumulative economic losses for the industrial sectors have been calculated for the recovery periods of different durations. The results show that: (1) the longer the period that an industrial department returns to normal output after haze pollution has impacted, the greater the cumulative economic loss will be; (2) when the recovery period is one year, the cumulative economic loss value computed by the dynamic EC + IO model is much smaller than the loss value obtained by the static EC + IO model; (3) the recovery curves of industrial sectors show that the recovery rate at the early stage is fast, while it is slow afterwards. Therefore, the governance work after the occurrence of haze pollution should be launched as soon as possible. This study provides a theoretical basis for evaluating the indirect economic losses of haze pollution and demonstrates the value of popularization and application.

## 1. Introduction

Outdoor air pollution, especially haze pollution, directly impacts the health of people and the transportation system, and indirectly affects other industrial departments. Currently, in China, the PM_2.5_ (particulate matter smaller than 2.5 μm) pollution, which is largely produced by exhaust emissions from motor vehicles, carries a serious threat to public health and the national economic system [1].

Outdoor air pollution (mainly PM_2.5_) leads to 3.3 million premature deaths per year worldwide, and surprisingly, 1.357 million, or 41.2 percent of the world’s total, occurred in China alone [2]. The death rate caused by atmospheric pollution in China is nearly one order of magnitude higher than that of road traffic injuries and AIDS (Acquired Immunodeficiency Syndrome), and air pollution is the main cause of death [3].

In contrast to the above studies on health effects, this study is to develop static and dynamic models that are capable of estimating indirect economic loss incurred by haze pollution for the Beijing area at refined industrial levels (particularly for the transportation and warehousing sector) and predicting the trend of the economic damage.

Indirect economic loss has been commonly recognized as consequential costs or the decline of direct economic loss [4]. It is normally discussed from the aspects of human and environmental impacts, which differs but is caused by direct loss [5].

Beijing is the capital of China, the world’s second most populous city (6336 square miles: urban 528 square miles and rural 5808 square miles) and the most populous capital city (21.7 million municipal population, 18.8 million urban, 24.0 million metro in 2017). The city, located in northern China, is governed as a direct-controlled municipality under the national government with sixteen urban, suburban, and rural districts.

For Beijing, the total haze pollution is determined as a combination of self-emitted and transmitted (drifted-in from outside) PM_2.5_ contamination. On the 14th May 2018, the Beijing Municipal Environmental Protection Bureau reported that about one third of annual PM_2.5_ in Beijing was contributed due to regional transmission from outside sources; whereas two thirds was due to local pollution emissions, of which 45% were due to mobile sources, including diesel vehicles, petrol vehicles, Beijing transit vehicles, transit diesel vehicles, aviation trains, non-road machinery, etc. [6]. Therefore, it is a critical and challenging to measure to estimate and even predict possible economic losses due to haze pollution.

### 1.1. Literature Survey

Haze pollution not only seriously affects people’s health but also incurs huge socio-economic losses [7,8,9]. The current research on the economic loss caused by haze pollution is mainly from the perspective of health. Ridkei [10] applied the human capital method to assess the economic loss caused by air pollution regarding various diseases and deaths in United States in 1958, and the results showed that the total health benefits of air pollution during the year was US $80.2 billion. This study was considered to be the beginning of health damage assessments in air pollution. Quah and Boon [11] studied the incidence, mortality and economic loss of airborne particulate matter (PM_10_) in Singapore, and they found that the total economic loss caused by air pollution in Singapore in 1999 accounted for 4.31% of the GDP (Gross Domestic Product) in that year. Yoo, Kwak and Lee [12] estimated that families with a 10% drop in the concentration of major pollutants in Seoul paid US $4.6 a month, and all Seoul residents had to pay US $169.5 million a year. Othman et al. [13] collected data on the daily hospitalization of 14 diseases related to haze pollution from four hospitals in Malaysia in 2005, 2006, 2008 and 2009 respectively, and assessed the health economic loss caused by haze pollution in the area. It was estimated that the average annual economic loss due to haze pollution was about US $91,000 for each hospitalized patient.

Research on the economic loss of air pollution in China began in the 1980s. Guo, Zhang and Li [14] comprehensively utilized the market value method, the opportunity cost method, the engineering cost method, the corrected human capital method and a large amount of statistics and testing data to assess China’s environmental pollution losses during the “6th Five-Year Plan” period (1981–1985). Results showed that the air pollution loss during this period was 12 billion RMB. Wang and Qu [15] used the exposure–response model and the production function method to quantify the loss of air pollution. It was found that the pollution loss of atmospheric resources in Jiangsu Province in the 1990s was as high as 10 billion RMB each year. Wan, Yang and Masui [16] estimated that the economic losses caused by particulate pollution based on the exposure–response model in Shanghai’s urban areas in 2001 were approximately US $625.4 million, which was 1.03% of the city’s GDP. Xia, Guan and Jiang [17] adopted the supply-driven input–output model to estimate the economic losses caused by the shortening of working hours in 2007 in 30 provinces in China related to air pollution, and the total economic loss was found to be 346.26 billion RMB (about 1.1% of the national GDP). This loss approximately equal to the annual GDP of Vietnam in 2010. Peng and Tian [18] used the vector autoregressive model (VAR model) to investigate the long-term dynamic characteristics of environmental pollution and economic growth in Hunan Province from 1985 to 2008, and found that economic growth was an important cause of environmental pollution. It has a reverse effect on economic growth and a certain hysteresis effect. Mu and Zhang [19] evaluated the direct economic loss in areas affected by haze pollution in China in 2013 and found the country’s direct economic loss on transportation and health was about 23 billion RMB through the integrated utilization of a direct loss assessment, disease cost method, human capital law and other methods. Wang et al. [20] used the changes in labor supply and additional medical expenses as the conduction variables to feed back into the computable general equilibrium model (CGE model). It was found that the extra medical expenses for PM_2.5_ pollution in Beijing in 2013 were about RMB 1.113 billion. The negative health effects resulted in a loss of about 23.396 billion RMB in the total output of the industrial sector, and a loss of about 901 million RMB in GDP. Hao et al. [21] used urban-level panel data from 2013 to 2015 to study the impact of PM_2.5_ concentration on urban per capita GDP, and introduced a series of time and regional models to control for the fixed effect. It was found that smog pollution did have a significant negative impact on economic development, and when other conditions were the same, each increase in PM_2.5_ concentration could result in a decrease in per capita GDP of about 2500 yuan. Fan and Wu [22] used Shanghai’s per capita GDP and three types of haze pollution indicators as research samples. Through the establishment of the VAR model, the long-term equilibrium relationship and dynamic impact mechanism between Shanghai’s economic growth and haze pollution levels were studied. It has been found that there is a two-way mechanism between Shanghai’s economic growth and haze pollution, and the adverse effect of haze pollution on economic growth is greater than the impact of economic growth on haze pollution.

### 1.2. The Scope and Methods of Our Study

Besides health consequences, the public are also concerned about the indirect economic losses caused by haze pollution on other aspects [23,24]. Particularly, in addition to short-term economic damages, the public prefer to pursue a long-term loss estimate on the economy, which involves indirect economic loss, a topic of this study.

Visibility is badly reduced during smog pollution, which seriously affects the transportation system, and often triggers ripple effects to other industrial sectors such as the food and tobacco sectors, damaging trade network links between regions and hence impacting the whole economic system. This potential indirect economic loss is more profound than the direct economic loss [25]. The value of indirect economic loss caused by haze pollution is studied in this paper, reflecting the indirect effects of smog pollution.

By including the indirect economic loss caused by haze pollution, this study combines an econometric model (EC model) with an input–output model (IO model) to extend the traditional input–output model, and so constructs the econometrics and input–output joint model (EC + IO model). Previous EC + IO models were mostly used for the study of industrial structures, rarely for air pollution loss.

At present, the human capital method, exposure–response model, and other methods are commonly employed to assess the economic loss of smog. These methods can only roughly calculate the total economic loss caused by haze pollution and cannot be refined to the local level [26,27,28]. A few studies have attempted to evaluate the economic losses of haze through traditional IO models and static CGE models. However, the traditional IO model is based on the premise of a linear relationship; whereas the static CGE model is very demanding on the data, and the entire process takes a long time and has a lower efficiency. An EC + IO model is established in this paper to randomize the deterministic IO model with the characteristics of the EC model, which can improve the ability of the traditional IO model to analyze problems and optimize the properties of the model. This proposed model preserves the decomposition of the traditional IO model sectors, and adds a random term in the EC model, making the model into a dynamic framework, which is able to not only reduce the static restrictions of the IO model, but also extend the linear constraints [29,30]. The EC + IO joint model absorbs the advantages of the IO model and the EC model, and it can reflect both the detailed departmental classification structure and randomness. At the same time, the EC + IO joint model connects the statistical yearbook with the data from the input–output table, which allows us to obtain relevant data from the input–output table containing non-edited year related data, making its data more integrated and expanding its data applications. The construction of dynamic EC + IO model can break the constraint on the time of the static EC + IO model and add the variable “recovery period” making the model more extensive and dynamic in practical forecasting. For these two models, data is easier to obtain and greatly reduces the time consumption.

Previously, a static EC + IO model for direct and indirect economic damages incurred by typhoon events for China was presented in 2013 [29]. It has been expanded to the study of damages caused by haze pollution for the Beijing area in 2013, with the method from a static EC + IO model to a dynamic one. In that case, conductive and interactive effects between sectors can be considered and further studied.

The remainder of this paper is organized as follows. Section 2 introduces the relevant models. In Section 3, the static and dynamic EC + IO models will be developed. In Section 4, for the Beijing area in 2013, an empirical analysis will be performed on the indirect economic losses for all 42 industrial sectors, which were caused, through ripple effects, by direct damage to the transportation and warehousing sector. Section 5 contains a discussion on the model’s limitations. Finally, the conclusion is drawn in Section 6.

## 2. The System Model

The input–output identity based on line relationship and pure industrial sector as follows:(1)$$\sum_{j=1}^{n}x_{ij}+Y_{i}=X_{i},$$where $x_{ij}$ is the number of products allocated by sector i to the sector j. $X_{i}$ is the total output of sector i. $Y_{i}$ is the end use of sector i. A is the direct consumption coefficient matrix whose element is $a_{ij}=\frac{x_{ij}}{X_{j}}$.

Equation (1) is expressed as a matrix:(2)AX+Y=X,

It is equivalent to:(3)$$X={\left(I-A\right)}^{-1}Y,$$where ${\left(I-A\right)}^{-1}$ is a Leontief inverse matrix.

The connection of the EC model and the IO model is usually done by using the end-use variable Y in the IO model. The Y in the non-supplemental regional input–output table consists of six components: household consumption, government consumption, total capital formation, net exports, domestic inflows outside the province, and domestic inflows outside the province. Due to the limitation of regional accounting data, when establishing the regional EC + IO model, only the resident consumption in the Y is considered to establish the EC model, i.e., to make the resident consumption data endogenous. Other end-use items are treated as exogenous variables and are substituted into the model using raw data. Using the results of the EC model to drive the IO model, the model built by the connection method is the EC + IO joint model.

## 3. Model Establishment

### 3.1. EC Model

Based on the regional non-supplemented input–output table, a regional EC + IO model of line relationship was constructed in this paper. Due to the limitation of regional accounting data, only the EC model of residential consumption was established. Based on the dual social structure in China, the urban residents’ consumption model and rural consumption model were established respectively in this paper. Time series data can be used to establish the following model:

Urban residents’ consumption equation:(4)$$LnC_{t}=\beta_{01}+\beta_{11}LnDIS_{t}+\beta_{21}LnC_{t-1}+\varepsilon .$$

Rural residents’ consumption equation:(5)$$LnC_{t}=\beta_{02}+\beta_{12}LnDIS_{t}+\beta_{22}LnC_{t-1}+\varepsilon .$$

Among them, in Equation (4), DIS is per capita disposable income, and in Equation (5), DIS per capita net income. $C_{t}$ is the residents’ consumption in the current period, $C_{t-1}$ denotes the residents’ consumption in the previous period. ε is a random item.

### 3.2. Static EC + IO Model

Using the prediction results of the EC model and the original data, the end-use part is denoted as, and Equation (3) becomes:(6)$$X^{1}={\left(I-A^{1}\right)}^{-1}Y^{1},$$where $A^{1}$ is the predicted direct consumption coefficient matrix, and its element is $a_{ij}^{1}=\frac{x_{ij}}{\sum_{i=1}^{n}x_{ij}+Y_{j}^{1}}$. Besides, $Y_{i}^{1}$ is the element of $Y^{1}$ and the element of $X^{1}$ is $X_{i}^{1}=\sum_{j=1}^{n}x_{ij}+Y_{i}^{1}$.

From Equation (6), the loss of the final product of the industrial sector can be regarded as the direct economic loss, i.e., $\Delta Y={\left(\Delta Y_{1}\Delta Y_{2},\ldots ,\Delta Y_{n}\right)}^{T}$. Then, the total product loss will be:(7)$$\Delta X={\left(I-A^{1}\right)}^{-1}\Delta Y,$$where ΔX is also considered as the total economic loss and the indirect economic loss is ΔX−ΔY.

In order to improve the accuracy of the assessment of indirect economic loss in each industrial sector, the total consumption coefficient will be used for the analysis [1]. It is obvious that the total consumption coefficient matrix can be regarded as the predicted total consumption coefficient matrix, and the relationship with the predicted direct consumption coefficient matrix is $B^{1}={\left(I-A^{1}\right)}^{-1}-I$. Equation (7) can then be transformed into:(8)$$\Delta X=(B^{1}+I)\Delta Y$$

Suppose that the loss in sector i is due to a certain pollution incident, and the final needs of other sectors remain unchanged, then the total output changing in all sectors is shown as follows:
(9)$$\left(\begin{array}{c} \Delta X_{1}\\ \Delta X_{2}\\ \ldots \\ \Delta X_{i}\\ \ldots \\ \Delta X_{n} \end{array}\right)=[\left(\begin{array}{cccccc} b_{11}^{1} & b_{12}^{1} & \ldots & b_{1i}^{1} & \ldots & b_{1n}^{1}\\ b_{21}^{1} & b_{22}^{1} & \ldots & b_{2i}^{1} & \ldots & b_{2n}^{1}\\ \ldots & \ldots & \ldots & \ldots & \ldots & \ldots \\ b_{i1}^{1} & b_{i2}^{1} & \ldots & b_{ii}^{1} & \ldots & b_{in}^{1}\\ \ldots & \ldots & \ldots & \ldots & \ldots & \ldots \\ b_{n1}^{1} & b_{n2}^{1} & \ldots & b_{ni}^{1} & \ldots & b_{nn}^{1} \end{array}\right)+\left(\begin{array}{cccccc} 1 & 0 & 0 & \ldots & 0 & 0\\ 0 & 1 & 0 & \ldots & 0 & 0\\ 0 & 0 & 1 & \ldots & 0 & 0\\ \ldots & \ldots & \ldots & \ldots & \ldots & \ldots \\ 0 & 0 & 0 & \ldots & 1 & 0\\ 0 & 0 & 0 & \ldots & 0 & 1 \end{array}\right)]\left(\begin{array}{c} 0\\ 0\\ \ldots \\ \Delta Y_{i}\\ \ldots \\ 0 \end{array}\right)=\left(\begin{array}{c} b_{1i}^{1}\Delta Y_{i}\\ b_{2i}^{1}\Delta Y_{i}\\ \ldots \\ b_{ii}^{1}\Delta Y_{i}\\ \ldots \\ b_{ni}^{1}\Delta Y_{i} \end{array}\right)+(\begin{array}{c} 0\\ 0\\ \ldots \\ \Delta Y_{i}\\ \ldots \\ 0 \end{array}),$$
where $b_{ij}^{1}(i,j=1,2,\ldots ,n)$ is the predicted total consumption factor, and the total economic loss for sector i is:(10)$$\Delta X_{i}=b_{ii}^{1}\Delta Y_{i}+\Delta Y_{i},$$where $\Delta Y_{i}$ is the direct economic loss of sector i and $b_{ii}\Delta Y_{i}$ is the indirect economic loss of the ith sector. The total economic loss for other sectors is:(11)$$\Delta X_{m}=b_{mi}^{1}\Delta Y_{i},m\neq i.$$

### 3.3. Dynamic EC + IO Model

The Leontief dynamic IO model is [31]:(12)X(t)−AX(t)−D[X(t+1)−X(t)]=U(t),where D is the investment coefficient matrix, D[X(t+1)−X(t)] is the productive investment matrix and U(t) is the final net demand matrix (D[X(t+1)−X(t)]+U(t)=Y(t)).

If let $Q=-D^{-1}$, then Equation (12) can be converted to:(13)X(t+1)−X(t)=Q[AX(t)+U(t)−X(t)].

The total economic loss ratio of sector i is $l_{i}=\Delta x_{i}/x_{i}$, where $\Delta x_{i}$ is the total economic loss of sector i. $x_{i}$ is the total output of sector i. $l_{i}$ is the element of matrix total loss proportion matrix L and matrix $L=X^{-1}\Delta X$. The loss ratio of demand for sector i is $u_{i}^{*}=\Delta u_{i}/x_{i}$, where $\Delta u_{i}$ is the demand loss for the i^th^ sector. $u_{i}^{*}$ is the element of the demand loss matrix $U^{*}$, then $U^{*}=X^{-1}\Delta U$. Equation (13) can be shown as:(14)$$l(t+1)-l(t)=Q[A^{*}l(t)+U^{*}(t)-l(t)],$$where $A^{*}=X^{-1}AX$.

The general solution of (14) is:(15)$$l(t)=l(0)e^{-Q(1-A^{*})t}+\int_{0}^{t}QU^{*}(s)e^{Q(1-A^{*})(s-t)}ds.$$

If the final requirements of the various industrial sectors remain the same, then $U^{*}=0$. Equation (15) becomes:(16)$$l(t)=l(0)e^{-Q(1-A^{*})t}.$$

When t→∞, the loss ratio l(t)→0 and the affected industrial sectors returned to normal. If only sector i is considered, Equation (16) can be changed to:(17)$$l_{i}(t)=l_{i}(0)e^{-q_{i}(1-q_{ii}^{*})t}.$$

The total economic loss $X_{i}(t)$ in sector i during the period of returning to normal production can be expressed as:(18)$$X_{i}(t)=x_{it}\int_{t=0}^{T}l_{i}(t)dt,$$where $x_{it}$ is the output of sector i during period t.

The dynamic EC + IO model predicts the end-use part of the unprocessed input–output table, denoted as $Y^{1}$. If the input–output inequality still holds, the total output of the $i^{th}$ sector is $X_{i}^{1}=\sum_{j=1}^{n}x_{ij}+Y_{i}^{1}$. As a result, the output of sector i during period t will change. Similarly, the rate of loss will change as well, and then Equation (17) becomes the following:(19)$$l_{i}{\left(t\right)}^{1}=l_{i}(0)e^{-q_{i}(1-q_{ii}^{*1})t},$$where $a_{ii}^{*1}$ is the element of matrix $A^{*1}=X^{-1}A^{1}X$, and $A^{1}$ is the predicted matrix of direct consumption coefficient.

The cumulative economic loss (i.e., the economic loss during the recovery period until the normal production is resumed) of sector i evaluated by the dynamic EC + IO model in period t during normal recovery is:(20)$$X_{i}(t)=x_{it}^{1}\int_{t=0}^{T}l_{i}{\left(t\right)}^{1}dt,$$where $X_{i}(t)$ is also considered as indirect economic loss and $x_{it}{}^{1}$ is the predicted output of sector i in one unit time.

## 4. Empirical Analysis

### 4.1. Data Sources

The relevant data from 1993–2012 in the Beijing Statistical Yearbook have been used to establish the EC model. In order to eliminate the effect of price changes, constant price (1993 = 100) data is required. The input–output table data comes from the 42 sector input–output table of Beijing in 2012, while the direct economic loss data is derived from the statistical study of Mu and Zhang [19].

### 4.2. Parameter Estimation of the EC Model

After taking the logarithm of the original data, it was found that all the indexes involved in the model were first-order single integer sequences. However, their first-order differential sequences have no unit root and are stable, thus meeting the cointegration preconditions. The long-term equilibrium relationship between the sequences of variables needs to be further studied. The E–G two-step method was used to test the cointegration relationship, and performs the ADF (Augmented Dickey-Fuller Test) unit root test on the regression residual e of the consumption equations of urban residents and rural residents respectively. The results are shown in Table 1.

From the table above, it can be seen that at the level of significance of 5%, the regression series ($e_{1}$ and $e_{2}$) of urban residents’ consumption and rural residents’ consumption are all stable. It is considered that there is a cointegration relationship among the variables, and the cointegration regression equations are as follows:

Urban residents’ consumption cointegration regression equation:(21)$$LnC_{t}=1.062+0.626LnDIS_{t}+0.244LnC_{t-1}.$$

Rural residents’ consumption cointegration regression equation:(22)$$LnC_{t}=0.294+0.914LnDIS_{t}+0.087LnC_{t-1}.$$

In Equation (21), the fact that ${\hat{R}}^{2}$ = 0.9975, the value of the F statistic is 3570.307, and the p value (α=5%) is far less than 0.05, which shows that the equation of urban residents’ consumption is significant. Similarly, in Equation (22), that fact that ${\hat{R}}^{2}$ = 0.9964, the value of the F statistic is 2513.692, and the p value (α=5%) is much less than 0.05, which shows that the equation of rural residents’ consumption is also significant.

### 4.3. Assessment of Indirect Economic Loss of Haze Pollution

#### 4.3.1. Forecast of Total Residential Consumption

Equations (21) and (22) can be used to predict the consumptions per capita for urban and rural residents living in Beijing in 2013, respectively. Multiplying these rates with the populations of urban and rural residents in that year, the total consumption of urban residents in Beijing in 2013 was 508.2 billion RMB, and the total consumption of rural residents was 3861.5 billion RMB. According to the input and output table for the 42 departments in Beijing in 2012, the percentage of residents’ consumption in each department was calculated. This percentage was taken as the percentage of residents’ consumption in various departments in 2013. Finally, the original data was used for other items of Y to obtain the total output of each department in 2013. The results are shown in Table 2.

#### 4.3.2. Empirical Results of Industrial Sector’s Indirect Economic Loss

##### Data Processing

The input–output table of the 42 industrial sectors in Beijing in 2012 was used in this paper, and data on the direct economic losses are from statistical research by Mu and Zhang [19]. In January 2013, the direct economic loss caused by haze pollution incidents in Beijing was 64.2 million RMB (1777.7 million RMB for haze-related health treatment, see Mu and Zhang [19]). The loss for the storage industry is regarded as the direct economic loss, under the transportation and warehousing sector.

##### Evaluation Results from Static EC + IO Model

By applying Equation (10), the total economic loss for the transportation and warehousing sector in Beijing in 2013 was estimated as 87.9 million RMB. This figure includes this sector’s direct economic loss (64.2 million RMB, See Mu and Zhang [19]) and indirect economic loss (23.7 million RMB, shown in Table 3 below). From Equation (11), the indirect economic losses of other sectors caused by the ripple effect triggered by the damage to the transportation and warehousing sector can be calculated, and the results are arranged in Table 3.

From Table 3, following the descending order of total losses in each sector, the top five industrial sectors impacted hardest by haze pollution in terms of value loss in Beijing in 2013 are the transportation and warehousing sector (23.7 million RMB), petroleum, coking products and processed nuclear fuel products sector (15.8 million RMB), finance sector (8.2 million RMB), oil and natural gas mining products sector (7.6 million RMB), and electricity, heat production and supply sector (6.9 million RMB). In addition, the top five industrial sectors affected by haze according to the indirect economic loss ratio are the metal products, machinery and equipment repair services sector (0.0715%), petroleum, coking products and processed nuclear fuel products sector (0.0138%), oil and gas mining products sector (0.0112%), transportation and warehousing sector (0.0069%), and textiles sector (0.0058%).

Moreover, it is noted that three sectors were ranked in the top five due to two indicators, i.e., the oil and natural gas mining products sector, petroleum, coking products and processed nuclear fuel products sector, and the transportation and warehousing sector. The total economic loss for these three industrial sectors was further calculated in Beijing in 2013, totaling 111.2 million RMB, which accounted for 66.91% of the total economic loss across all industrial sectors (166.2 million RMB). Therefore, these three sectors are the hardest hit by haze pollution.

##### Evaluation Results from Dynamic EC + IO Model

From the above, metal products, machinery and equipment repair services sector suffered from haze pollution in Beijing 2013 the highest indirect economic loss ratio. Hence, the dynamic EC + IO model was decided to use this sector to evaluate the cumulative economic loss during the recovery period. The indirect economic loss ratio of this sector is 0.0715%, i.e., $l^{1}(0)=0.000715$. Assuming that the industry returns to 99.99% of its original output after 30 days, i.e., l(30)=0.0001. From Equation (19), this sector’s q value (q=0.0691) was obtained, and when the industrial sector returns to normal output, the recovery equation becomes:(23)$$1-l(t)=1-0.000715\times e^{-0.0691\times t}.$$

The 30-day recovery curve for this sector is shown in Figure 1.

From the figure above, this sector’s recovery is faster in the early recovery period and slower in the later recovery period.

It can be seen from Equation (20) that when the recovery period of this sector continues for 30 days, the cumulative economic loss of the sector will be 125.0 thousand RMB. Similarly, the results of the economic loss can be calculated when the recovery periods vary from 10 days, 20 days, 30 days 60 days, 365 days, 5 years, 10 years and 20 years, which is shown in Table 4.

From the results in Table 4, it is seen that the shorter the recovery period, the smaller the cumulative economic loss caused by haze pollution to the department. The longer the recovery period, the larger the cumulative economic loss. When the recovery period is one year, the cumulative loss value obtained from the dynamic EC + IO is much smaller than that obtained by the static EC + IO model, because the loss rate in the dynamic EC + IO model gradually decreases over time while the static model uses an instantaneous loss rate as the constant annual loss rate, resulting in the estimated industrial economic loss being larger.

The specific recovery period for the metal products, machinery and equipment repair service sector was not studied after being hit by haze pollution. With regard to the recovery periods of this and other sectors, Hsiang and Jina’s extensive meta-analysis in 2014 about some 6700 metrological events that occurred from 1950 to 2008, found strong evidence that, after being hit by metrological disasters, the recovery periods for industrial sectors might last for an extremely long time and could vary from some years to even two decades.

## 5. Discussion

In this paper, the static and dynamic EC + IO models were established to estimate the indirect economic loss caused by haze pollution in the Beijing area. The application of the traditional IO model, which is based on the input–output table and updated every five years, results in a time difference between the data in the input–output table and the data in the statistical yearbook. The EC + IO model not only absorbs the advantages of the IO model and the EC model, but also connects the data in the statistical yearbook with the data in the input–output table to some extent. It is therefore more comprehensive than either of them due to obtaining relevant data of the input–output table that is to be in the incoming yearbook. When constructing the dynamic EC + IO model, adding the recovery period variables in this paper made the dynamic EC + IO model more widely used in practical applications. Therefore, compared with the traditional IO model, the EC + IO model not only expands its data application field, but also improves the accuracy of the evaluation results.

Study of the indirect economic impact of haze pollution in Beijing focuses on the assessment of indirect economic loss. Motor vehicle exhaust is one of the culprits of smog pollution. Visibility is reduced during haze pollution, which seriously affects the transportation system. The transportation industrial sector is damaged due to the correlation effect between industrial sectors, which will affect other sectors and the entire economic system. This potential indirect economic loss is far more profound than the impact of direct economic loss; compare with Gu’s recent result [7]. Gu applied the traditional IO model to assess the economic loss in 2013 caused by haze pollution in Beijing and found that the indirect economic loss caused by haze pollution amounted to 26.8 million RMB for the transportation and warehousing sector alone, which also resulted in a total indirect economic loss of 145.6 million RMB for transportation-related industrial sectors due to ripple effects. This result is higher than the static assessment of this paper (102.0 million RMB). The reason for this difference is that Gu adopted the 2010 Beijing input–output extension table to estimate the economic loss for 2013. Therefore, the estimates in this paper are more in line with the actual situation.

Through the dynamic EC + IO model, it is found in this study that the shorter the recovery period, the smaller the cumulative economic loss, whereas the longer the recovery period, the greater the cumulative economic loss. When the recovery period is one year, the loss rate gradually decreases with time, so the cumulative economic loss value is much smaller than the loss value obtained by the static model. In other words, the static EC + IO model overestimates the cumulative economic loss because it assumes a constant loss rate for a given period.

The results supported our team’s earlier research [1,29]. The EC + IO model is mostly used in the study of the industrial structure [32,33,34]. A further study that some scholars have established is an EC + IO model that predicts future emissions of atmospheric pollutants to reflect regional changes in emissions and economic structures in Chicago [34]. For now, this is the first time that the EC + IO model has been employed for assessing regional haze-associated indirect economic loss. Noticeably, the implementation of the dynamic EC + IO model in this paper eliminates (or reduces) the drawback of overestimation by the static EC + IO model. Hence, findings of this paper provide policy makers a better understanding and more insights into haze pollution, along with its impact on the economy. In addition, there is only little difference between the regional and national input–output tables, and hence the assessment process in this paper can also be promoted in other regions, and even the whole country.

The EC + IO model was established through the end-use variables in the IO model. This variable consists of six parts: household consumption, government consumption, total capital formation, net exports, domestic inflows outside the province, and domestic inflows outside the province. Due to the limitation of regional accounting data, only the EC model on resident consumption was established, and hence the proposed EC model may be further improved.

The EC + IO models can subdivide the distribution of economic loss by haze pollution, but this can only estimate industry-related losses, and cannot obtain a total value of economic loss by haze pollution. Moreover, when using the EC + IO model to calculate the associated economic loss, it is assumed that the final product of one or more severely impacted sectors have changed, while the final products of other sectors remain unchanged, and there is suspicion of underestimating economic loss from haze pollution.

## 6. Conclusions

Due to the complexity of the economic system, the preparation of the input–output table is undoubtedly a complicated project, generally compiled once every five years in China. Hence, input–output tables are often not continuous in time, and because of the classification of different industrial sectors, an input and output table with the annual preparation of the statistical yearbook cannot be a good choice. In this study, the EC + IO model makes it possible to combine the uncertainty of the real economy with a detailed departmental classification structure. The EC model is used to predict the relevant parts of the input–output table, to obtain the input–output tables of the unedited years, and therefore to expand the input–output model. The EC + IO model has been utilized to quantitatively assess the indirect economic loss caused by haze pollution. The results indicate that:With regards to the economic loss results, according to the static EC + IO model, the indirect economic loss caused by major haze pollution events in Beijing in 2013 was 23.7 million RMB for the transportation and warehousing sector alone; the total indirect economic loss due to the ripple effect triggered by damage to the transportation and warehousing sector was 102.0 million RMB. The value of this loss is large and cannot be ignored. Economic development can be one of the approaches used to reduce haze pollution.The three sectors most affected by haze pollution are: (1) the oil and natural gas mining products sector; (2) the petroleum, coking products and processed nuclear fuel products sector; and (3) the transportation and warehousing sector. When haze pollution occurs, the relevant government departments should pay special attention to these industrial sectors.From the departmental recovery curve for the dynamic EC + IO model, a sector’s recovery initially goes faster but becomes slower later in the period. The longer a sector recovers, the bigger its losses become. Therefore, after the occurrence of haze pollution, the relevant government departments should start governance work as soon as possible, in order to reduce the sector’s recovery duration, so that the affected industrial sectors can resume normal output.According to the evaluation results of the dynamic EC + IO model, it can be believed that the longer the recovery period is, the greater the economic loss will be. Hence, the relevant departments should consider the length of the recovery period to develop haze pollution governance policy.

The loss rate in the dynamic EC + IO model decreases over time, implying that the static EC + IO model overestimates the cumulative economic loss since the static EC + IO model assumes a constant loss rate for a given period. The result of the dynamic EC + IO model is more in line with the actual situation.

## Figures and Tables

**Figure 1 ijerph-16-02328-f001:**
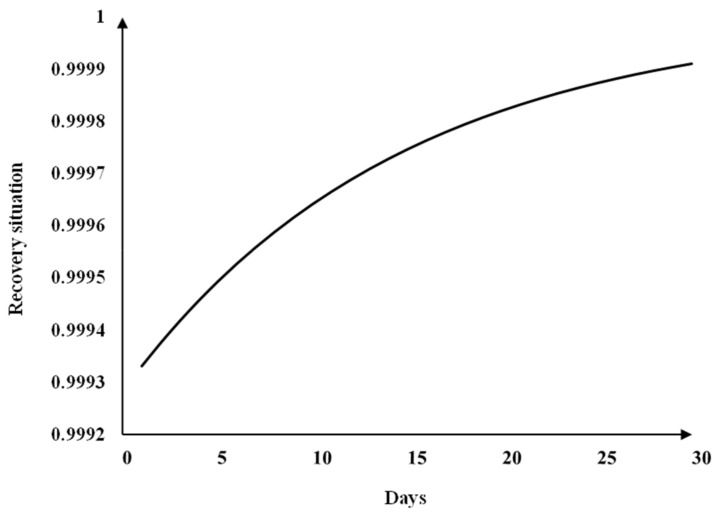
Thirty day metal products, machinery and equipment repair service sector recovery curve.

**Table 1 ijerph-16-02328-t001:** ADF unit root test results for regression residual sequences.

Variables	T Statistics	1% Threshold	5% Threshold	Test Form (c,t,k)	*p*-Value	Conclusion (*α* = 0.05)
$e_{1}$	−3.569	−2.699	−1.961	(0,0,0)	0.0013	stationary
$e_{2}$	−2.900	−2.699	−1.961	(0,0,0)	0.0063	stationary

Note: $e_{1}$ represents the regression sequence of urban residents’ consumption, $e_{2}$ is rural the residents’. ADF: Augmented Dickey-Fuller Test.

**Table 2 ijerph-16-02328-t002:** Predicted total output for each industrial sector in 2013.

No.	Industrial Sector	Total Output/Million RMB
1	Agriculture, forestry, animal husbandry and fishery	47,706.7
2	Coal mining industry	118,782.9
3	Oil and natural gas mining products	67,610.5
4	Metal mining industry	29,972.1
5	Non-metallic minerals and other mining industry	30,240.6
6	Food and tobacco	120,672.5
7	Textile industry	8105.2
8	Textile clothing footwear leather down and its products	30,975.5
9	Woodworking and furniture	13,779.1
10	Paper printing and cultural and educational sporting goods	52,469.5
11	Petroleum, coking and processed nuclear fuel products	114,225.7
12	chemical product	180,805.2
13	Non-metallic mineral products	60,885.0
14	Metal smelting and calendering products	88,774.5
15	Metal products industry	45,225.9
16	General Equipment	78,925.2
17	Professional setting	67,906.8
18	Transportation equipment	362,258.0
19	Electrical machinery and equipment	92,811.3
20	Communications equipment, computers and other	274,597.0
21	Instrumentation	28,891.9
22	Other manufacturing products	10,834.5
23	Waste scrap	2404.1
24	Metal Products, Machinery and Equipment Repair	5043.9
25	Electricity, heat production and supply	362,264.0
26	Gas production and supply	23,652.1
27	Water production and supply	5813.7
28	Construction industry	440,748.7
29	Wholesale and Retail	430,978.5
30	Transportation and warehousing	341,639.8
31	Accommodation and dining	127,355.5
32	Information transmission, software and information	333,324.7
33	Finance	414,562.4
34	Real estate	219,714.6
35	Leasing and business services	247,071.4
36	Scientific research and technical services	386,633.2
37	Water conservancy, environment and public	32,244.3
38	Residents services, repairs and other services	29,791.8
39	Education	122,793.0
40	Health and social work	115,255.6
41	Culture, sports and entertainment	115,273.1
42	Public administration, social security and social organization	157,876.3

**Table 3 ijerph-16-02328-t003:** Indirect economic loss for each industrial sector in 2013.

No.	Industrial Sector	Loss Value/Million RMB
1	Transportation and warehousing	23.7
2	Petroleum, coking and processed nuclear fuel products	15.8
3	Finance	8.2
4	Oil and natural gas mining products	7.6
5	Electricity, heat production and supply	6.9
6	Leasing and business services	5.0
7	Wholesale and Retail	4.1
8	Metal Products, Machinery and Equipment Repair	3.6
9	Transportation equipment	2.7
10	Metal smelting and calendering products	2.2
11	Chemical products industry	2.1
12	Paper printing and cultural and educational sporting goods	2.0
13	Communications equipment, computers and others	1.9
14	General Equipment	1.4
15	Food and tobacco	1.4
16	Accommodation and dining	1.3
17	Information transmission, software and information	1.2
18	Scientific research and technical services	1.2
19	Real estate	1.1
20	Residents services, repairs and other services	1.1
21	Coal mining industry	0.9
22	Electrical machinery and equipment	0.8
23	Metal products industry	0.7
24	Gas production and supply	0.7
25	Agriculture, forestry, animal husbandry and fishery	0.6
26	Textile industry	0.5
27	Non-metallic mineral products	0.5
28	Construction industry	0.5
29	Instrumentation	0.4
30	Textile clothing footwear leather down and its products	0.3
31	Woodworking and furniture	0.3
32	Professional setting	0.3
33	Culture, sports and entertainment	0.3
34	Education	0.2
35	Non-metallic minerals and other mining industry	0.1
36	Other manufacturing products	0.1
37	Waste scrap	0.1
38	Water production and supply	0.1
39	Public administration, social security and social organ	0.05
40	Water conservancy, environment and public facilities management	0.04
41	Metal mining industry	0.02
42	Health and social work	0.01
Total indirect economic loss		102.0

**Table 4 ijerph-16-02328-t004:** Cumulative economic loss value in different recovery periods.

Recovery Period *	Industrial Economic System Recovery Ratio	Cumulative Economic Loss/Million RMB
10 days	0.2073	0.04
20 days	0.1036	0.08
30 days	0.0691	0.13
60 days	0.0345	0.25
365 days	0.0057	1.52
5 years	0.0011	9.08
10 years	0.00057	15.76
20 years	0.00027	33.25

* See remark below.

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
