# Peer review of "Indirect Economic Impact Incurred by Haze Pollution: An Econometric and Input–Output Joint Model"

_ijerph, 2019, doi:10.3390/ijerph16132328_

Round 1
Reviewer 1 Report
Dear Authors,
This manuscript is very interesting and a useful paper.
It would be imagined authors worked very hard.
Do you have a plan to confirm whether your model is reasonable every year?
What is the best way to use this model on economy and deal in countries?
Author Response
First, thank you and your reviewers for the extremely helpful comments provided for our paper. In the revision of the paper we have addressed all comments. The following is a listing of the comments made and our responses detailing how we addressed the issues raised by the reviewer during the review process.
Reviewer 1 Comments to the Author
Comment #1: Do you have a plan to confirm whether your model is reasonable every year?
Response to Comment #1: Thank you for your comment. We are looking forward to making that happen. Haze pollution monitoring and warning would be a long-term process; while continuous confirmation helps us to improve our model as well.
Comment #2: What is the best way to use this model on economy and deal in countries?
Response to Comment #2: Thank you for your comment. Different countries have variable economic situation and conditions. Cross-country deal would be an interesting and challenging topic and yet be covered by this paper. We may think of this in our future efforts.
Yours Faithfully,
Keyao Chen
Reviewer 2 Report
This paper is generally well written, interesting and has important scientific context. It should be of great interest to the readers. However, there are language and grammatical errors which the author should send the paper for professional editing. Additional clarifications are needed in some sections.
My first concern is the introduction section. It lacked depth and needs to be enriched. The author made a lot of statements without proper references or citations (For example: Line 34-37). In addition, the paper did not make an introduction about the model which will be presented in the article. For example, the problems and questions were missing which must be answered by the paper. In addition, please give examples on “Indirect economic loss”.
About the system model, the authors illustrated 6 components but only 1 component (resident consumption) is considered in the EC model. First of all, is resident consumption equivalent to the household consumption as described among the 6 components? Second, what was the “limitation of the regional accounting data”? How would the limitation affect the statistical outcome? Perhaps more thorough justifications and clarifications would be needed, as the unclear of the parameter design would affect the understanding of the statistical results.
The discussion and conclusions can be improved. The reviewer see the paper as a solid source of data and study, but there are difficulty to see the policy implications and also the implications of significance. An expanded explanation or enrichment of what significance looks like on the ground would be helpful.
Other issues needing attention:
Table: Table 3 are not revealing. In line 289 – 303, it mentioned the top 5 sectors but were not clearly identified in Table 3. Authors may consider to highlight the sectors in the table or rank the sectors according to the loss. In addition, what is the “numbering” column means in Table 2 and 3?
Consistency: Please be careful with the consistency of the acronyms, for example “EC+IO model” vs “EC + IO model”
Equation 4 and 5: The reviewer has trouble understanding the difference with “x” and without “x” in equation 4 and 5. Is that a typo?
Line 253 – 259: Please correct the equation numbers in the equations (Should Line 254 be Equation 22?) and also within the script.
Overall, it is an interesting study, and should be considered for publication, once the above comments have been addressed.
Author Response
First, thank you and your reviewers for the extremely helpful comments provided for our paper. In the revision of the paper we have addressed all comments. The following is a listing of the comments made and our responses detailing how we addressed the issues raised by the reviewer during the review process.
Reviewer 2 Comments to the Author
Comment #1: The author made a lot of statements without proper references or citations (For example: Line 34-37).
Response to Comment #1: Thank you for your comment. More references have been cited in the Introduction section. For example, a new reference has been cited at Line 37, as shown below:
“Outdoor air pollution, especially haze pollution, directly impact the health of people and the transportation system, and indirectly affect other industrial departments. At present in China, the PM2.5 pollution, which is largely produced by exhaust emissions from motor vehicles, carries a serious threat to the public health and the national economic system [1].” ( explained in Line 37 )
Comment #2: the paper did not make an introduction about the model which will be presented in the article. For example, the problems and questions were missing which must be answered by the paper.
Response to Comment #2: Thank you for your comment. We have revised the Introduction section by bridging the contents to help audience understand the problems caused by haze pollution and the issues that we are trying to solve. ( explained in Line 60-61,153-154 )
.
Comment #3: please give examples on “Indirect economic loss”.
Response to Comment #3: The term “Indirect economic loss” has been explained in Line 46-48, as shown below:
“In terms of Indirect Economic Loss, it has been commonly recognized as consequential costs or decline of direct economic loss [4]. It is normally discussed from the aspects of human and environmental impacts, which differs but is caused by direct loss [5].”
Comment #4: the authors illustrated 6 components but only 1 component (resident consumption) is considered in the EC model.
Response to Comment #4: Resident consumption is considered as the typical dataset to verify the model. Other components were not included in the paper due to confidentiality of corresponding data from local government.
Comment #5: The discussion and conclusions can be improved.
Response to Comment #5: More contents have been added in the sections. (explained in Line 407,412,418-421 )
Comment #6: Table 3 are not revealing. In line 289 – 303, it mentioned the top 5 sectors but were not clearly identified in Table 3. Authors may consider to highlight the sectors in the table or rank the sectors according to the loss.
Response to Comment #6: More contents have been added to explain Table 3. Meanwhile, the loss value of the top five sectors have been highlighted in Table 3. (explained in Line 294-295 )
Comment #7: what is the “numbering” column means in Table 2 and 3?
Response to Comment #7: Numbering is used for the sequence of sector items for sorting purposes.
Comment #8: Please be careful with the consistency of the acronyms, for example “EC+IO model” vs “EC + IO model”.
Response to Comment #8: Thank you for your suggestion. We have checked all acronyms, including “EC + IO model”, to make them consistent.
Comment #9: Equation 4 and 5: The reviewer has trouble understanding the difference with “x” and without “x” in equation 4 and 5. Is that a typo?
Response to Comment #9: Thank you for your comment. They are typos and we have corrected them.
Comment #10: Line 253 – 259: Please correct the equation numbers in the equations (Should Line 254 be Equation 22?) and also within the script.
Response to Comment #10: Thank you for the correction. We have updated corresponding contents and the equation.
Yours Faithfully,
Keyao Chen